# Using Confounded Data in Offline RL

**Maxime Gasse**
ServiceNow Research, Montréal QC, Canada
`maxime.gasse@servicenow.com`

**Damien Grasset**
IRT Saint Exupéry, Montréal QC, Canada
`damien.grasset@irt-saintexupery.com`

**Guillaume Gaudron**
Ubisoft La Forge, Bordeaux, France
`guillaume.gaudron@ubisoft.com`

**Pierre-Yves Oudeyer**
Inria Bordeaux Sud-Ouest, Bordeaux, France
`pierre-yves.oudeyer@inria.fr`

## Abstract

In this work we consider the problem of confounding in offline RL, also referred to as the delusion problem [Ortega et al., 2021]. While it is known that learning from purely offline data is a hazardous endeavor in the presence of confounding, in this paper we show that offline, confounded data can be safely combined with online, non-confounded data to improve the sample-efficiency of model-based RL. We import ideas from the well-established framework of $do$-calculus to express model-based RL as a causal inference problem, thus bridging the fields of RL and causality. We propose a latent-based method which we prove is correct and efficient, in the sense that it attains better generalization guarantees thanks to the offline, confounded data (in the asymptotic case), regardless of the expert's behavior. We illustrate the effectiveness of our method on a series of synthetic experiments.

## 1 Introduction

As human beings, understanding cause and effect is crucial to successfully navigate our environment. This is also true of reinforcement learning (RL) agents, and in particular model-based agents, who must learn the effects of their own actions (interventions) on the environment. From this perspective, offline RL is analogous to the problem of causal inference from observational data, which requires assumptions about the data-generating process [1]. In the context of Markov Decision Processes (MDPs), estimating causal effects from offline data can be shown to be straightforward, due to the absence of confounding. In Partially-Observable MDPs (POMDPs) however, it requires the additional assumption that the data-collection agent did not use any privileged information besides that available to the learning agent. Relaxing this assumption results in confounded data, which off-the-shelf offline RL algorithms [Lange et al., 2012, Levine et al., 2020] should not use.

A typical example is in the context of medicine, when offline data is collected from physicians who may rely on information absent from their patient's medical records, such as their wealthiness or their lifestyle. Suppose that wealthy patients in general get prescribed specific treatments by their physicians, because they can afford it, while being less at risk to develop severe conditions regardless of their treatment, because they can also afford a healthier lifestyle. This creates a spurious correlation called confounding, and will cause a naive recommender system to wrongly infer that costly treatments have positive health effects. Another example is in the context of autonomous driving, when offline data is collected from human drivers who have a wider field of vision than the camera on which the robot driver relies. Suppose human drivers push the brakes when they see a person waiting to cross the street, and only when the person walks in front of the car it enters the

---

[1] "behind every causal conclusion there must lie some causal assumption that is not testable in observational studies", Pearl [2009b].

camera's field of vision. Then, again, a naive robot might wrongly infer that whenever the brakes are pushed, a person appears in front of the car. In order to minimize collisions with pedestrians, it might get regrettably reluctant to push the brakes.

Of course, in both those situations, the learning agent can infer the right causal model by disregarding the (confounded) offline data altogether, and by relying solely on online data instead, collected from its own interactions. However, in both those situations also, performing interventions for the sole purpose of seeing what happens is impractical, while collecting offline data by observing the behaviour of human agents is much more affordable. The question we address in this paper is, how can confounded, offline data be leveraged by an RL agent who can also collect online data?

To answer this question we import tools and ideas from the well-established field of causality [Pearl, 2009a] into the model-based RL framework. We formalize model-based RL as a causal inference problem using the framework of $do$-calculus [Pearl, 2012], and we present a generic method for combining online and offline data in model-based RL, with a formal proof of correctness and efficiency even in the presence of confounding. We propose a practical implementation in the tabular setting, and present three experiments on synthetic toy problems that illustrate its effectiveness.

## 2 Background

### 2.1 Notation

In this paper, upper-case letters in italics denote random variables (e.g. $X, Y$), while their lower-case counterpart denote their value (e.g. $x, y$) and their calligraphic counterpart their domain (e.g., $x \in \mathcal{X}$). For simplicity we consider only discrete random variables. To keep our notation uncluttered, with a slight abuse of notation we use $p(x)$ to denote sometimes the event probability $p(X = x)$, and sometimes the whole probability distribution of $X$, which should be clear from the context. In sequential models we also distinguish random variables with a temporal index $t$, which might be fixed (e.g., $o_0, o_1$ ), or undefined (e.g., $p(s_{t+1}|s_t, a_t)$ denotes at the same time the distributions $p(s_1|s_0, a_0)$ and $p(s_2|s_1, a_1)$). We also adopt a compact notation for sequences of contiguous variables (e.g., $s_{0 \to T} = (s_0, \dots, s_T) \in \mathcal{S}^{T+1}$ ), and for summations over sets ($\sum_{x \in \mathcal{X}} \iff \sum_x^{\mathcal{X}}$). We assume the reader is familiar with the concepts of conditional independence ($X \perp\!\!\!\perp Y \mid Z$) and probabilistic graphical models based on directed acyclic graphs (DAGs), which can be found in most introductory textbooks, e.g. Pearl [1989], Studeny [2005], Koller and Friedman [2009].

### 2.2 Partially-Observable Markov Decision Process

We consider episodic Partially-Observable Markov Decision Processes (POMDPs) of the form $M = (\mathcal{S}, \mathcal{O}, \mathcal{A}, p_{init}, p_{trans}, p_{obs}, r)$, with hidden states $s \in \mathcal{S}$, observations $o \in \mathcal{O}$, actions $a \in \mathcal{A}$, initial and transition state distributions $p_{init}(s_0)$ and $p_{trans}(s_{t+1}|s_t, a_t)$, observation distribution $p_{obs}(o_t|s_t)$, and reward function $r : \mathcal{O} \to \mathbb{R}$. For simplicity we assume episodes $\tau = (o_0, a_0, \dots, o_T)$ of finite length $|\tau| = T$, and we introduce the concept of a history at time $t$, $h_t = (o_0, a_0, \dots, o_t)$. The control mechanism is represented as a stochastic policy $\pi$, which together with the POMDP dynamics $p_{init}$, $p_{trans}$ and $p_{obs}$ defines a probability distribution over trajectories, $p(\tau)$. In this work we consider two types of control policies $\pi$, which result in two distinct data-generation regimes.

**Definition 2.1** (Standard POMDP regime). In the *standard POMDP regime*, actions are decided based only on the available information from the past, $H_t$, according to a *standard policy* $\pi_{std}(a_t|h_t)$. This results in the data-generation process depicted in Figure 1, and trajectory distribution

$$p_{std}(\tau) = \sum_{s_{0 \to |\tau|}}^{\mathcal{S}^{|\tau|+1}} p_{init}(s_0)p_{obs}(o_0|s_0) \prod_{t=0}^{|\tau|-1} \pi_{std}(a_t|h_t)p_{trans}(s_{t+1}|s_t, a_t)p_{obs}(o_{t+1}|s_{t+1}).$$

This standard regime is that of the regular POMDP control problem, which formulates as:

$$\pi_{std}^{\star} = \arg\max_{\pi_{std}} \mathbb{E}_{\tau \sim p_{std}} \left[ \sum_{t=0}^{|\tau|} r(o_t) \right]. \tag{1}$$

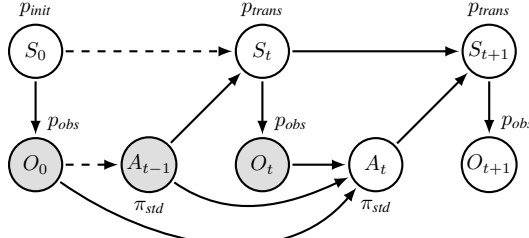

Figure 1: The standard POMDP setting (no confounding).

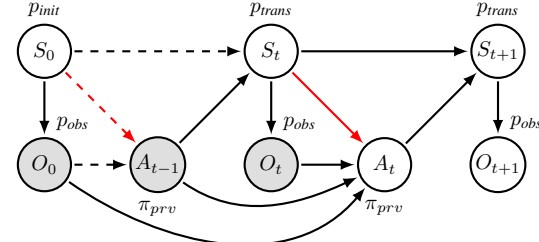

Figure 2: The privileged POMDP setting (confounding).

**Definition 2.2** (Privileged POMDP regime). In the *privileged POMDP regime*, actions can be decided based on the hidden state $S_t$ as well, according to a *privileged policy* $\pi_{std}(a_t|h_t, s_t)$. This results in the data-generation process depicted in Figure 2, with trajectory distribution

$$p_{prv}(\tau) = \sum_{s_{0\to|\tau|}}^{\mathcal{S}^{|\tau|+1}} p_{init}(s_0) p_{obs}(o_0|s_0) \prod_{t=0}^{|\tau|-1} \pi_{prv}(a_t|h_t, s_t) p_{trans}(s_{t+1}|s_t, a_t) p_{obs}(o_{t+1}|s_{t+1}).$$

This privileged regime allows us to consider situations where trajectories are collected by observing an external agent who uses privileged information, in the extreme case the entire POMDP hidden state. Such a privileged agent can be for example a human driver in the context of autonomous driving, who might have access to privileged information that the learning robot doesn't have for driving, such as the weather forecast. There lies the origin of the confounding problem in offline RL.

## 2.3   Causality and do-calculus

Several frameworks exist in the literature for reasoning about causality [Pearl, 2009a, Imbens and Rubin, 2015, Dawid, 2021]. Here we follow the framework of Judea Pearl, whose concept of *ladder of causation* is particularly relevant to answer RL questions. The first level of the ladder, *association*, relates to the passive observation of an external agent acting in the environment, while the second level, *intervention*, relates to the question of what will happen to the environment as a result of the observer's own actions. The tool of *do*-calculus [Pearl, 2012], presented in Appendix A, acts as a bridge between these two levels, and is typically used to answer whether and interventional distribution, such as $p(y|do(x), z)$, can be identified from an observational distribution, such as $p(x, y, z)$. In a nutshell, in causal systems that can be expressed as DAGs, an intervention $do(x)$ forces the variables in $X$ to take the specific value $X = x$ regardless of their causal ancestors in the graph, and queries of form $p(y|do(x), z)$ measure the effect of an intervention $do(X = x)$ on an outcome event $Y = y$, in the context where another event $Z = z$ is also observed. In this paper, we will use $do$-calculus to reason formally about offline model-based RL in different POMDP data-collection regimes, which entail different causal graphs.

## 3   Model-based RL as causal inference

Decision-making problems are inherently causal [Gershman, 2017, Dawid, 2021]. In POMDPs, model-based RL relies on measuring the causal effect of immediate interventions, $do(a_t)$, on the next observation, $o_{t+1}$, given that past observations, $o_{0\to t}$, and past interventions, $do(a_{0\to t-1})$, have already happened. Such causal queries are embodied in the causal transition model $p(o_{t+1}|o_{0\to t}, do(a_{0\to t}))^2$, which depends only on the POMDP dynamics in $M$, and not on the control policy $\pi$. Together with the initial distribution $p(o_0)$, this causal model allows for the evaluation of any standard control policy $\pi_{std}(a_t|h_t)$. Model-based RL then decomposes Equation (1) into two sub-problems:

1. learning: given a dataset $\mathcal{D}$, estimate a model $\hat{q}(o_{t+1}|h_t, a_t) \approx p(o_{t+1}|o_{0\to t}, do(a_{0\to t}))$;

---

[2]Such a notation can be found also in [Ortega et al., 2021]

2. planning: given a history $h_t$ and the model $\hat{q}$, derive an optimal action $a_t$.

In this work we consider only the first problem, that is, learning the causal transition model from data. Next, we show using *do*-calculus that this problem can be either trivial or impossible, depending on whether the data is collected using a standard or a privileged control policy.

## 3.1 In the standard POMDP regime

In the standard POMDP regime, we assume access to a dataset $\mathcal{D}_{std} \sim p_{std}(\tau)$ of episodes $\tau$ collected using an arbitrary standard policy $\pi_{std}(a_t|h_t)$. A key characteristic in this setting is that $A_t \perp\!\!\!\perp S_t \mid H_t$ is always true, that is, every action is independent of the current hidden state given the current history. By applying *do*-calculus on the causal graph from Figure 1, the causal model can be shown to be trivially identifiable as

$$p(o_{t+1}|o_{0\to t}, do(a_{0\to t})) = p_{std}(o_{t+1}|h_t, a_t).$$

Because of this property, any trajectory $\tau \sim p_{std}(\tau)$ can be interpreted as an *interventional* trajectory, where the learning agent itself could have decided on each of the action $a_t$ in $\tau$. Thus, in the remainder of the paper we will interchangeably call the standard POMDP regime the *interventional regime*, and any dataset $\mathcal{D}_{std}$ collected in this regime an *interventional dataset*.

Assuming sufficient exploration, which is achieved if the control policy is strictly positive ($\pi_{std}(a_t|h_t) > 0, \forall a_t, h_t$), an unbiased estimator of the POMDP causal model can be obtained from $\mathcal{D}_{std}$ via log-likelihood maximization,

$$\hat{q} = \arg\max_{q \in \mathcal{Q}} \sum_{\tau}^{\mathcal{D}_{std}} \sum_{t=0}^{|\tau|-1} \log q(o_{t+1}|h_t, a_t). \tag{2}$$

This corresponds to the simplest and most common form of model learning via supervised learning [Moerland et al., 2020], which effectively solves our causal inference problem.

## 3.2 In the privileged POMDP regime

In the privileged POMDP regime, we assume access to a dataset $\mathcal{D}_{prv} \sim p_{prv}(\tau)$ of episodes $\tau$ collected using an arbitrary privileged policy $\pi_{prv}(a_t|h_t, s_t)$. In this setting, actions might not be independent of the current hidden state given the current history, and thus $A_t \perp\!\!\!\perp S_t \mid H_t$ might not hold. Because each hidden state $S_t$ both has a causal effect on the current action $A_t$ and the next observation $O_{t+1}$, it acts as a hidden confounder in the POMDP causal transition model. This confounding effect can not be adjusted for without observing the hidden states of the POMDP, and applying *do*-calculus on the causal graph from Figure 2 results in the causal model $p(o_{t+1}|o_{0\to t}, do(a_{0\to t}))$ being non-identifiable from $p_{prv}(\tau)$. In particular,

$$p(o_{t+1}|o_{0\to t}, do(a_{0\to t})) \neq p_{prv}(o_{t+1}|h_t, a_t).$$

Because of this, trajectories $\tau \sim p_{prv}(\tau)$ cannot be interpreted as interventional. To better relate to the causality literature, we will interchangeably call the privileged POMDP regime the *observational regime*, and any dataset $\mathcal{D}_{prv}$ collected in this regime an *observational dataset*.

Note that, as a consequence of this non-identifiability, naively applying any off-the-shelf offline RL algorithm [Lange et al., 2012, Levine et al., 2020] on an observational dataset such as $\mathcal{D}_{prv}$ is a risky endeavour, and might result in biased transition models and value functions, and sub-optimal policies.

## 3.3 Connection to online and offline RL

To relate the concepts of standard (interventional) and privileged (observational) data to online and offline RL, the key question to ask is, when the samples were collected, could the control policy have used privileged information besides the history $h_t$? Or, more formally, can we guarantee that $A_t \perp\!\!\!\perp S_t \mid H_t$ did hold in the data-generating process?

**In online RL**, the learning agent explicitly controls the data-collection policy, so by design it can not rely on privileged information, hence $A_t \perp\!\!\!\perp S_t \mid H_t$ always holds. Therefore, data collected in an online RL setting can be safely treated as interventional, and the causal transition model can be directly estimated using Equation (2).

**In offline RL**, the learning agent might have limited knowledge about the data-collection policy, sometimes no knowledge at all. In some settings, if it can be shown that the policy could not have used any privileged information, then the offline data can be treated as interventional. For example, with human replays from Atari video games, it is hard to imagine a human player having access to more information from the machine's internal state than the regular video and audio outputs from the game. But in more general offline RL settings, access to privileged information can not be dismissed. This is particularly true with human demonstrations collected in the wild, such as in the context of autonomous driving, medical recommender systems (examples in Section 1), or question answering systems Ortega et al. [2021]. In that case, the offline trajectories can not be considered interventional, and the offline dataset must be treated as observational.

## 4 Combining observational and interventional data

Given enough online data, RL agents can learn optimal policies. But in some situations collecting a large online (interventional) dataset can be expensive (recording a robot driver in the wild), while collecting a large offline (observational) dataset from demonstrations is relatively cheap (recording human drivers in the wild). Is it possible then to leverage such offline data to improve the sample-efficiency of an online RL agent, even in the presence of confounding? [3]

### 4.1 Problem statement

We consider two datasets of POMDP trajectories, $\mathcal{D}_{std}$ and $\mathcal{D}_{prv}$, sampled respectively in the standard (interventional) and the privileged (observational) regime. We then ask the following question: can the observational dataset $\mathcal{D}_{prv}$ be used in combination to the interventional dataset $\mathcal{D}_{std}$, to improve the POMDP causal transition model $p(o_{t+1}|o_{0\rightarrow t}, do(a_{0\rightarrow t}))$ that would be obtained from Equation (2) using the interventional data only? As we will see, answering thing question will require to go beyond the identifiability framework of $do$-calculus.

### 4.2 The augmented learning procedure

Since both datasets are sampled from the same POMDP ($p_{init}, p_{trans}, p_{obs}$) controlled in different ways, we introduce a regime indicator [Dawid, 2021] variable $I \in \{0, 1\}$ so that $\mathcal{D}_{prv} \sim p(\tau|i = 0)$ and $\mathcal{D}_{std} \sim p(\tau|i = 1)$, with the augmented control policy $\pi(a_t|h_t, s_t, i) = \pi_{prv}(a_t|h_t, s_t)$ when $i = 0$ and $\pi_{std}(a_t|h_t)$ when $i = 1$. This results in the augmented data-generating process from Figure 3.

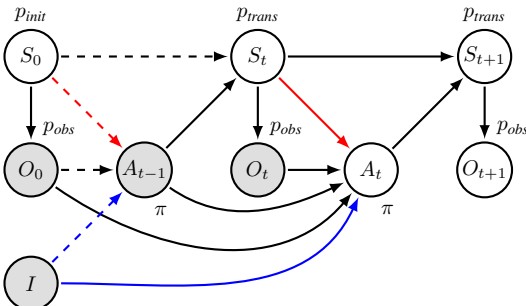

Figure 3: Augmented POMDP setting, with a policy regime indicator $I$ taking values in $\{0, 1\}$ (1=standard policy, 0=privileged policy), such that $\pi(a_t|h_t, s_t, i = 1) = \pi(a_t|h_t, i = 1)$, which enforces the contextual independence $A_t \perp\!\!\!\perp S_t \mid H_t, I = 1$.

In order to learn the causal transition model $p(o_{t+1}|o_{0\rightarrow t}, do(a_{0\rightarrow t}))$ we propose the following two-step procedure, which relies on fitting a latent probabilistic model $\hat{q}$ that explains both $\mathcal{D}_{std}$ and $\mathcal{D}_{prv}$. Our latent model is constrained to respect the structure of our augmented POMDP, with a latent variable $z_t \in \mathcal{Z}$ to substitute the hidden state $s_t \in \mathcal{S}$.

---

[3]Note that we consider this question in its broadest, without further assumptions about the observed offline agent. The offline agent might act sub-optimally, or optimally according to a different reward function than the learning agent.

**Learning** Our learning problem formulates as standard likelihood maximization,

$$\hat{q} = \arg\max_{q \in \mathcal{Q}} \sum_{(\tau)}^{\mathcal{D}_{prv}} \log q(\tau | i = 0) + \sum_{(\tau)}^{\mathcal{D}_{std}} \log q(\tau | i = 1), \tag{3}$$

with $\mathcal{Q}$ the family of sequential models that respect the augmented POMDP structural constraints,

$$q(\tau | i = 0) = \sum_{z_{0 \to |\tau|}}^{\mathcal{Z}^{|\tau|+1}} q_{init}(z_0) q_{obs}(o_0 | z_0) \prod_{t=0}^{|\tau|-1} q_{prv}(a_t | h_t, z_t) q_{trans}(z_{t+1} | a_t, z_t) q_{obs}(o_{t+1} | z_{t+1}), \text{ and}$$

$$q(\tau | i = 1) = \sum_{z_{0 \to |\tau|}}^{\mathcal{Z}^{|\tau|+1}} q_{init}(z_0) q_{obs}(o_0 | z_0) \prod_{t=0}^{|\tau|-1} q_{std}(a_t | h_t) q_{trans}(z_{t+1} | a_t, z_t) q_{obs}(o_{t+1} | z_{t+1}).$$

The latent model $q(\tau|i)$ decomposes into five components, $q_{init}$, $q_{trans}$, $q_{obs}$, $q_{std}$ and $q_{prv}$. In practice however, the component $q_{std}(a_t|h_t)$ (standard policy model) does not affect the recovered causal transition model, and does not have to be learned. Likewise, the component $q_{prv}(a_t|h_t, z_t)$ (privileged policy model) can be substituted for $q_{prv}(a_t|z_t)$ with no consequence, which is simpler to implement. As a result, only four components have to be learned: the initial latent model $q_{init}(z_0)$, the observation model $q_{obs}(o_t|z_t)$, the latent transition model $q_{trans}(z_{t+1}|z_t, a_t)$ and the privileged agent's behaviour model $q_{prv}(a_t|z_t)$. Each of these can be implemented using a feed-forward neural network, or a probability table in the discrete case. Interestingly, in the pure interventional regime when $\mathcal{D}_{prv} = \emptyset$, our augmented learning problem (3) boils down to solving (2) with a latent-based model.

**Inference** We recover the causal transition model $\hat{q}(o_{t+1}|o_{0 \to t}, do(a_{0 \to t})) = \hat{q}(o_{t+1}|h_t, a_t, i = 1)$ by applying $do$-calculus on the augmented DAG from Figure 3, with $z_t$ instead of $s_t$. The procedure conveniently unrolls as a forward algorithm at test time, and relies on the recurrent computation of $\hat{q}(z_t|h_t, i = 1)$, a.k.a. the agent's belief state at time $t$ Cassandra [1998], Striebel [1965]. First, the initial belief state at $t = 0$ is recovered as

$$\hat{q}(z_0|h_0, i = 1) = \frac{\hat{q}_{init}(z_0)\hat{q}_{obs}(o_0|z_0)}{\sum_{z_0}^{\mathcal{Z}} \hat{q}_{init}(z_0)\hat{q}_{obs}(o_0|z_0)}.$$

Then, for every $0 \leq t < T$, the causal transition model is recovered as

$$\hat{q}(z_{t+1}, o_{t+1}|h_t, a_t, i = 1) = \sum_{z_t}^{\mathcal{Z}} \hat{q}(z_t|h_t, i = 1)\hat{q}_{trans}(z_{t+1}|z_t, a_t)\hat{q}_{obs}(o_{t+1}|z_{t+1}),$$

$$\hat{q}(o_{t+1}|h_t, a_t, i = 1) = \sum_{z_{t+1}}^{\mathcal{Z}} \hat{q}(z_{t+1}, o_{t+1}|h_t, a_t, i = 1),$$

and the next belief state is updated to

$$\hat{q}(z_{t+1}|h_{t+1}, i = 1) = \frac{\hat{q}(z_{t+1}, o_{t+1}|h_t, a_t, i = 1)}{\sum_{z_{t+1}}^{\mathcal{Z}} \hat{q}(z_{t+1}, o_{t+1}|h_t, a_t, i = 1)}.$$

How does the observational data $\mathcal{D}_{prv}$ influence the causal transition model $\hat{q}(o_{t+1}|o_{0 \to t}, do(a_{0 \to t}))$? The intuition is as follows. The learned model $\hat{q}$ must fit both observational and interventional data by sharing the same latent variables $Z_t$, and the same building blocs $\hat{q}_{init}(z_0)$, $\hat{q}_{obs}(o_t|z_t)$ and $\hat{q}_{trans}(z_{t+1}|z_t, a_t)$. The privileged agent behaviour model, $\hat{q}_{prv}(a_t|z_t)$, is the only component that can allow for discrepancies between the two regimes, and it offers a limited flexibility. As a result, the observational distribution $\hat{q}(\tau|i = 0)$ estimated from $\mathcal{D}_{prv}$ can be seen as an unbiased regularizer for the interventional distribution $\hat{q}(\tau|i = 1)$ estimated from $\mathcal{D}_{std}$. This regularization helps prevent overfitting when learning from limited interventional data $\mathcal{D}_{std}$, and improves the generalization performance of the estimated causal transition model.

### 4.3 Theoretical guarantees

In this section we show that our two-step approach is 1) correct, in the sense that it yields an unbiased estimator of the standard POMDP causal transition model and 2) efficient, in the sense that it yields a better estimator than the one based on interventional data only (asymptotically in the number of

observational data). All proofs are deferred to Appendix C, and a companion example is given in Appendix B.

First we show that the recovered estimator is unbiased, and then we derive bounds for $\hat{q}(o_{t+1}|o_{0\to t}, do(a_{0\to t}))$ in the asymptotic observational scenario $|\mathcal{D}_{prv}| \to \infty$ (regardless of the interventional data $\mathcal{D}_{std}$).

**Proposition 4.1.** *Assuming* $|\mathcal{Z}| \geq |\mathcal{S}|$, $\hat{q}(o_{t+1}|o_{0\to t}, do(a_{0\to t}))$ *is an unbiased estimator of* $p(o_{t+1}|o_{0\to t}, do(a_{0\to t}))$.

**Theorem 4.2.** *Assuming* $|\mathcal{D}_{prv}| \to \infty$, *for any* $\mathcal{D}_{std}$ *the recovered causal model is bounded as follows:*

$$\prod_{t=0}^{T-1} \hat{q}(o_{t+1}|o_{0\to t}, do(a_{0\to t})) \geq \prod_{t=0}^{T-1} p(a_t|h_t, i=0)p(o_{t+1}|h_t, a_t, i=0), \text{ and}$$

$$\prod_{t=0}^{T-1} \hat{q}(o_{t+1}|o_{0\to t}, do(a_{0\to t})) \leq \prod_{t=0}^{T-1} p(a_t|h_t, i=0)p(o_{t+1}|h_t, a_t, i=0) + 1 - \prod_{t=0}^{T-1} p(a_t|h_t, i=0),$$

$\forall h_{T-1}, a_{T-1}, T \geq 1$ *where* $p(h_{T-1}, a_{T-1}, i=0) > 0$.

Note that Theorem 4.2 generalizes a famous results in econometrics known as Manski's bounds Manski [1990], which corresponds to the setting where $T = 1$. As a direct consequence, in the asymptotic case, using observational data ensures stronger generalization guarantees for the recovered causal transition model than using no observational data.

**Corollary 4.3.** *For any* $\mathcal{D}_{int}$, *the estimator* $\hat{q}(o_{t+1}|o_{0\to t}, do(a_{0\to t}))$ *recovered after solving (3) with* $|\mathcal{D}_{obs}| \to \infty$ *offers strictly better generalization guarantees than the one with* $|\mathcal{D}_{obs}| = 0$.

## 4.4 Related work

**Causal RL** A whole body of work exists around the question of merging interventional and observational data in RL in the presence of confounding. Bareinboim et al. [2015] study a sequential decision problem similar to ours, but assume that expert intentions are observed both in the interventional and the observational regimes, i.e., prior to doing interventions the learning agent can ask "what would the expert do in my situation?" This artificially introduces an intermediate, observed variable $\hat{a}_t = f(o_t)$ with the property that $p_{prv}(a_t = \hat{a}_t|\hat{a}_t) = 1$, which effectively removes any confounding ($A_t \perp\!\!\!\perp S_t|H_t$). Zhang and Bareinboim [2017, 2021] relax this assumption in the context of binary bandits, and later on in the more general context of dynamic treatment regimes [Zhang and Bareinboim, 2019, 2020]. They derive causal bounds similar to ours (Theorem 4.2), and propose a two-step approach which first extracts causal bounds from observational data, and then uses these bounds in an online RL algorithm. While their method nicely tackles the question of leveraging observational data for online exploration, it does not account for uncertainty in the bounds estimated from the observational data. In comparison, our latent-based approach is more flexible, as it never computes explicit bounds, but rather lets the learning agent decide through (3) how data from both regimes influence the final transition model, depending of the number of samples available. Kallus et al. [2018] also propose a two-step learning procedure to combine observational and interventional data in the context of binary contextual bandits, which relies on a series of strong parametric assumptions (strong one-way overlap, linearity, non-singularity etc.). Finally, a specific instantiation of this framework is off-policy evaluation, i.e., estimating the performance of a policy $\pi$ using observational data only, which corresponds to the specific setting $|\mathcal{D}_{int}| = 0$. While it can be shown that the causal transition model is in general not recoverable in the presence of confounding, a growing body of literature still tries to tackle this challenge by introducing additional structural or parametric assumptions on the data-generating process [Lu et al., 2018, Tennenholtz et al., 2020, Bennett et al., 2021].

**Large sequence models** A recent trend in RL is to apply large sequence models to estimate the environment's dynamics in a model-based fashion Schrittwieser et al. [2021], Janner et al. [2021], or to parameterize a goal-conditioned policy in a model-free fashion Chen et al. [2021], Zheng et al. [2022]. While large sequence models appear promising for efficiently combining offline and online data, they remain vulnerable to confounding, as pinpointed by Ortega et al. [2021]. Because it follows a generic model-based approach, our method could be easily combined with a large sequence model to address large-scale RL scenarios, while being robust to confounding.

# 5 Experiments

Given two datasets of standard (interventional) and privileged (observational) POMDP trajectories, $\mathcal{D}_{std}$ and $\mathcal{D}_{prv}$, our *augmented* method consists in recovering a causal model of the POMDP dynamics by solving Equation (3), and then extracting $\hat{q}(o_{t+1}|o_{0\to t}, do(a_{0\to t}))$ from $\hat{q}_{init}$, $\hat{q}_{trans}$ and $\hat{q}_{obs}$. To answer the question we asked in Section 4, we compare our method against two baseline variants: *no obs* which discards the observational dataset, and solves Equation (3) with $\mathcal{D}_{prv} \leftarrow \emptyset$, and *naive* which naively combines the observational and interventional datasets as if there was no confounding, and solves Equation (3) with $\mathcal{D}_{std} \leftarrow \mathcal{D}_{std} \cup \mathcal{D}_{prv}$ and $\mathcal{D}_{prv} \leftarrow \emptyset$.

## 5.1 Experimental setup

We train all three model-based methods, *augmented*, *no obs* and *naive*, using the same model architecture and training procedure. Each building bloc $\hat{q}_{init}$, $\hat{q}_{trans}$, $\hat{q}_{obs}$ and $\hat{q}_{prv}$ consists in a tabular logistic model, and Equation (3) is solved via mini-batch stochastic gradient descent using Adam [Kingma and Ba, 2015]. Once the POMDP dynamics are recovered we extract $\hat{q}(o_0)$ and $\hat{q}(o_{t+1}|o_{0\to t}, do(a_{0\to t}))$ to train a "dreamer" agent [Hafner et al., 2021] via actor-critic, implemented as a feed-forward neural network that takes as input the recovered POMDP belief state, $\hat{q}(s_t|o_{0\to t}, do(a_{0\to t-1}))$. Finally, we evaluate both the quality of the causal transition model $\hat{q}(o_{t+1}|o_{0\to t}, do(a_{0\to t}))$, in terms of the Jensen-Shannon divergence to $p(o_{t+1}|o_{0\to t}, do(a_{0\to t}))$, and the performance of the resulting RL agent, in terms of its cumulated reward, in the real test environment.

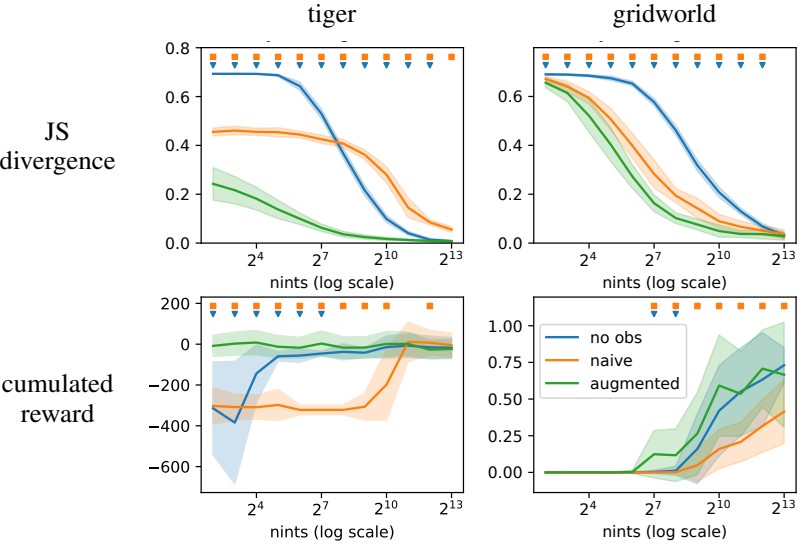

Figure 4: **Top**: quality of the transition model obtained by each method, in terms of the Jensen-Shannon divergence to $p(o_{t+1}|o_{0\to t}, do(a_{0\to t}))$ (lower the better). **Bottom**: performance obtained after training an RL agent, in terms of cumulative reward (higher the better). We report the mean and the standard deviation over 20 seeds. Little markers indicate a significant difference between our *augmented* method and the baselines *no obs* (down triangles) and *naive* (squares), using a two-sided Wilcoxon signed-rank test with $\alpha < 5\%$ [Demsar, 2006].

We conduct experiments on two synthetic toy problems, *tiger* and *sloppy gridworld*. In each scenario we sample observational (privileged) trajectories from a policy that relies on the POMDP's internal state, while we collect interventional (standard) trajectories using random exploration. Each time we measure the effect of combining a large, fixed observational dataset $\mathcal{D}_{prv}$ of size 8192 ($2^{13}$) with a growing interventional dataset $\mathcal{D}_{std}$, of size ranging from 4 ($2^2$) to 8192 ($2^{13}$) on a logarithmic scale. We repeat each experiment 20 times with different random seeds to account for variability.

Our complete experimental details are available in Appendix D, and the code to reproduce our experiments is available at `https://github.com/gasse/causal-rl`. We present additional experiments in Appendix E where we investigate robustness to different types of confounding (privileged policy), along with a third experiment in the simple binary bandit setting.

## 5.2 Detailed results

**Tiger** is a classic, small-scale POMDP from Cassandra et al. [1994] with $|\mathcal{S}| = 6$ hidden states and a time horizon $T = 50$. The learning agent's observation is a noisy information about the tiger's position (roar perceived left or right), and its actions are either to listen again or to open a door (left or right), which triggers a +10 (treasure) or -100 (tiger) reward and resets the tiger to a random position. The privileged agent has full knowledge of the tiger's position, and acts as follows

| | *action* | | |
|---|---|---|---|
| *tiger's position* | listen again | open left door | open right door |
| left | 0.05 | 0.3 | 0.65 |
| right | 0.05 | 0.8 | 0.15 |

Privileged policy $\pi_{prv}(action|tiger's\ position)$

As can be seen in Figure 4, this privileged policy results in confounding in the observational dataset $\mathcal{D}_{prv}$, which hurts the *naive* method. The *naive* method requires $|\mathcal{D}_{std}| = 2^{11}$ interventional trajectories to overcome the confounding and obtain a good policy. The *no obs* method, which does not use the observational data at all, obtains a good policy earlier with only $|\mathcal{D}_{std}| = 2^5$ interventional trajectories. Our *augmented* method, thanks to its correct use of the observational data, obtains a good policy even earlier, using only $|\mathcal{D}_{std}| = 2^2$ interventional trajectories.

**Sloppy gridworld** is inspired from Alt et al. [2020], and constitutes a more challenging POMDP with $|\mathcal{S}| = 21$ hidden states and a time horizon $T = 20$. Here the agent starts on the top-left corner of a small 5x5 grid, and tries to reach a target placed behind a large wall at the bottom side. The grid is sloppy, meaning that the environment will execute the agent's actions, *top*, *right*, *bottom*, *left* or *idle*, with 50% chances, and will execute a random action otherwise. The privileged agent has full knowledge of its position at each time step, and uses a shortest-path algorithm to decide on its next action. The learning agent is only revealed its position with 20% chances at each time step, and is blind otherwise (a dummy position is revealed). Here again, from Figure 4 it is clear that the *naive* method is hurt due to the confounding, while our *augmented* method benefits from the observational data compared to the *no obs* method. In Figure 5 we focus on the $|\mathcal{D}_{std}| = 128\ (2^7)$ trajectories mark, with a heat-map of the test-time trajectories resulting from each method. At this point, only our *augmented* method manages to cross the wall and reach the target, while the two other methods still struggle to escape the initial position.

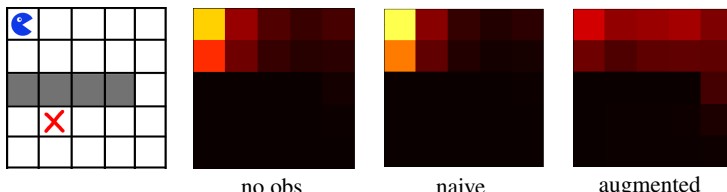

no obs      naive      augmented

Figure 5: Focus on the sloppy gridworld experiment at the $|\mathcal{D}_{std}| = 2^7$ mark. **Left**: the initial grid. **Right**: a heatmap of the tiles visited by the RL agents at test time. At this point, only the *augmented* method has learned how to pass the wall.

## 6 Conclusion

In this paper we have presented a simple, generic method for combining interventional and observational (potentially confounded) data in model-based reinforcement learning for POMDPs. We have demonstrated that our method is correct and efficient in the asymptotic case (infinite observational data), and we have illustrated its effectiveness on two synthetic toy problems. A future direction is to investigate this method in the high-dimensional POMDP setting, where learning a latent-based transition model is more challenging. We hope that this work will help bridge the gap between the fields of RL and causality, and will convince the RL community that causality is an adequate tool to reason about observational data, which is plentiful in the world.

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

# Appendices

## A    Introduction to do-calculus

The framework of $do$-calculus [Pearl, 2012] was proposed as an intuitive tool to answer identifiability questions given a causal graph $\mathcal{G}$, such as, can the interventional distribution $p(y|do(x), z)$ be recovered from the observational distributions $p(y, x, z)$? Do-calculus relies on three graphical rules, which depend solely on the existence of specific structural constraints in $G$:

- R1: insertion/deletion of observations, $p(y|do(x), z, w) = p(y|do(x), w)$ if $Y$ and $Z$ are $d$-separated by $X \cup W$ in $\mathcal{G}^{\star}$, the graph obtained from $\mathcal{G}$ by removing all arrows pointing into variables in $X$.

- R2: action/observation exchange, $p(y|do(x), do(z), w) = p(y|do(x), z, w)$ if $Y$ and $Z$ are $d$-separated by $X \cup W$ in $\mathcal{G}^{\dagger}$, the graph obtained from $\mathcal{G}$ by removing all arrows pointing into variables in $X$ and all arrows pointing out of variables in $Z$.

- R3: insertion/deletion of actions, $p(y|do(x), do(z), w) = p(y|do(x), w)$ if $Y$ and $Z$ are $d$-separated by $X \cup W$ in $\mathcal{G}^{\ddagger}$, the graph obtained from $\mathcal{G}$ by first removing all the arrows pointing into variables in $X$ (thus creating $\mathcal{G}^{\star}$) and then removing all of the arrows pointing into variables in $Z$ that are not ancestors of any variable in $W$ in $\mathcal{G}^{\star}$.

This set of rules has been shown to be complete [Huang and Valtorta, 2006, Shpitser and Pearl, 2006], and results in an algorithm polynomial in the number of nodes in $\mathcal{G}$ to answer identifiability questions, which either outputs "no" or "yes" along with an estimate (a recovery formula) based on observational quantities. We refer the reader to Pearl [2012] for a thorough introduction to $do$-calculus.

## B    Companion example: the door problem

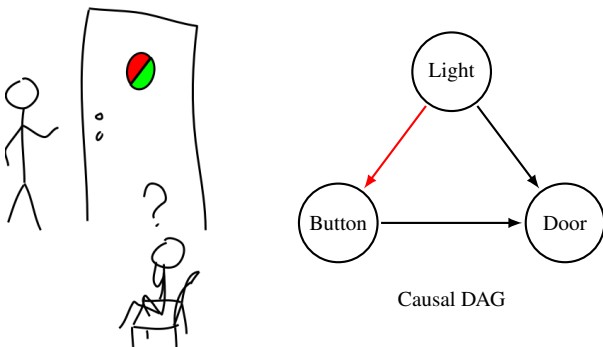

Figure 6: The door problem. You are sitting in a room with a door, a light that can be red or green, and two buttons that will open the door depending on the light color. You can collect data samples in two ways, either from interventions, i.e., you get up and press the buttons (expensive), or from observations, i.e., you watch someone else press the buttons (cheap). A key detail: you're colorblind and can't distinguish red from green. Your goal is to find which button is more likely to open the door.

Consider the door problem described in Figure 6. The mechanism responsible for opening the door works as follows: when the light is red, button A opens the door, when the light is green, button B opens the door. The light is red 60% of the time, and green the rest of the time. I am told nothing about the door's mechanism, except that it depends on both the light color and the button pressed ($Light \rightarrow Door \leftarrow Button$). Since I am colorblind I cannot use the light color to make decisions, and the question I am interested in is simply, which button is more likely to open the door? In the do-calculus framework, this question translates to

$$\arg\max_{button \in \{A, B\}} \; p(door{=}open|do(button)).$$

I thus need to estimate two causal queries: $p(door{=}open|do(button = A))$ and $p(door{=}open|do(button = B))$.

**Interventional setting** If I observe my own (or another colorblind person's) interactions with the door, then I know that which button is pressed is unrelated to which color the light is (*Light $\nrightarrow$ Button*). Then, I can directly estimate the causal effect of the button on the door,

$$p(door\text{=}open|do(button)) = p_{std}(door\text{=}open|button).$$

In this regime, regardless of which policy is used to collect (*button, door*) samples, eventually I realize that button A has more chances of opening the door (60%) than button B (40%), and thus is the optimal action[4].

**Observational setting** Assume now that I observe another person interacting with the door. I do not know whether that person is colorblind or not (*Light $\rightarrow$ Button* is possible). In this regime, without additional knowledge, I cannot recover the causal queries $p(door\text{=}open|do(button))$ from the observed distribution $p(door, button)$. In the *do*-calculus framework, the queries are said *non identifiable*. For example, with a privileged policy $\pi_{prv}(button\text{=}A|light\text{=}red) = 0.9$ and $\pi_{prv}(button\text{=}A|light\text{=}green) = 0.4$ the observed transition probabilities are $p_{prv}(door\text{=}open|button = A) = 0.77$ and $p_{prv}(door\text{=}open|button = B) = 0.8$, which are not causal due to confounding.

**Augmented setting** First, consider that I can observe many (*button, door*) interactions from a non-colorblind person, who's policy is $\pi(button\text{=}A|light\text{=}red) = 0.9$ and $\pi(button\text{=}A|light\text{=}green) = 0.4$. Then I can safely infer from Theorem 4.2 that $p(door\text{=}open|do(button\text{=}A)) \in [0.54, 0.84]$ and $p(door\text{=}open|do(button\text{=}B)) \in [0.24, 0.94]$. Then, I get a chance to interact with the door, and I decide to press $A$ 10 times and $B$ 10 times. I am unlucky, and my interventional study results in the following frequencies: $q(door\text{=}open|do(button\text{=}A)) = 0.5$ and $q(door\text{=}open|do(button\text{=}B)) = 0.5$. This does not coincide with my (reliable) observational study, and therefore I adjust $q(door\text{=}open|do(button\text{=}A))$ to its lower bound $0.54$. I now believe that pressing $A$ is more likely to be my optimal strategy.

## C  Proofs.

**Proposition 4.1.** *Assuming $|\mathcal{Z}| \geq |\mathcal{S}|$, $\hat{q}(o_{t+1}|o_{0\rightarrow t}, do(a_{0\rightarrow t}))$ is an unbiased estimator of $p(o_{t+1}|o_{0\rightarrow t}, do(a_{0\rightarrow t}))$.*

*Proof.* The proof is straightforward. First, we have that $\mathcal{D} \sim p(\tau, i)$. Second, we have $p \in \mathcal{Q}$, because $\mathcal{Q}$ is only restricted to the augmented POMDP constraints, and because its latent space is sufficiently large ($|\mathcal{Z}| \geq |\mathcal{S}|$). Therefore, $\hat{q}(\tau, i)$ solution of (3) is an unbiased estimator of $p(\tau, i)$, and in particular $\hat{q}(o_{t+1}|h_t, a_t, i = 1)$ is an unbiased estimator of $p(o_{t+1}|h_t, a_t, i = 1)$. $\square$

**Theorem 4.2.** *Assuming $|\mathcal{D}_{prv}| \rightarrow \infty$, for any $\mathcal{D}_{std}$ the recovered causal model is bounded as follows:*

$$\prod_{t=0}^{T-1} \hat{q}(o_{t+1}|o_{0\rightarrow t}, do(a_{0\rightarrow t})) \geq \prod_{t=0}^{T-1} p(a_t|h_t, i = 0)p(o_{t+1}|h_t, a_t, i = 0), \text{ and}$$

$$\prod_{t=0}^{T-1} \hat{q}(o_{t+1}|o_{0\rightarrow t}, do(a_{0\rightarrow t})) \leq \prod_{t=0}^{T-1} p(a_t|h_t, i = 0)p(o_{t+1}|h_t, a_t, i = 0) + 1 - \prod_{t=0}^{T-1} p(a_t|h_t, i = 0),$$

$\forall h_{T-1}, a_{T-1}, T \geq 1$ *where* $p(h_{T-1}, a_{T-1}, i = 0) > 0$.

*Proof of Theorem 4.2.* Consider $q(\tau, i) \in \mathcal{Q}$ any distribution that follows our augmented POMDP constraints. Then, for every $T \geq 1$ we have

$$\prod_{t=0}^{T-1} q(a_t|h_t, i)q(o_{t+1}|h_t, a_t, i) = \frac{q(\tau|i)}{q(h_0|i)}$$

$$= \sum_{z_{0\rightarrow T}}^{\mathcal{Z}^{T+1}} q(z_0|h_0, i) \prod_{t=0}^{T-1} q(a_t, z_{t+1}, o_{t+1}|z_t, h_t, i),$$

---

[4]One assumption though is strict positivity, $\pi(button) > 0 \ \forall button$, which ensures that both buttons are pressed.

by using $A_t, Z_{t+1}, O_{t+1} \perp\!\!\!\perp Z_{0 \to t-1} \mid Z_t, H_t, I$, which can be read via $d$-separation in the augmented POMDP DAG. Likewise, for every $t \geq 0$ we have

$$q(o_{t+1}|h_t, a_t, i = 1) = \sum_{z_{t+1}}^{\mathcal{Z}} q(z_{t+1}, o_{t+1}|h_t, a_t, i = 1)$$

$$= \sum_{z_t}^{\mathcal{Z}} q(z_t|h_t, i = 1) \sum_{z_{t+1}}^{\mathcal{Z}} q(z_{t+1}, o_{t+1}|z_t, h_t, a_t, i = 0),$$

by using $Z_t \perp\!\!\!\perp A_t \mid H_t, I = 1$ and $Z_{t+1}, O_{t+1} \perp\!\!\!\perp I \mid Z_t, A_t, H_t$. Then for every $t \geq 1$ we can further write

$$q(o_{t+1}|h_t, a_t, i = 1) = \sum_{z_t}^{\mathcal{Z}} \frac{q(z_t, o_t|h_{t-1}, a_{t-1}, i = 1)}{q(o_t|h_{t-1}, a_{t-1}, i = 1)} \sum_{z_{t+1}}^{\mathcal{Z}} q(z_{t+1}, o_{t+1}|z_t, h_t, a_t, i = 0).$$

By recursively decomposing every $q(z_t, o_t|h_{t-1}, a_{t-1}, i = 1)$ until $t = 0$, and finally by using $Z_0 \perp\!\!\!\perp I \mid H_0$, we obtain that for any $T \geq 1$

$$\prod_{t=0}^{T-1} q(o_{t+1}|h_t, a_t, i = 1) = \sum_{z_{0 \to T}}^{\mathcal{Z}^{T+1}} q(z_0|h_0, i = 0) \prod_{t=0}^{T-1} q(z_{t+1}, o_{t+1}|z_t, a_t, h_t, i = 0),$$

which can be re-expressed as

$$\prod_{t=0}^{T-1} q(o_{t+1}|h_t, a_t, i = 1) = \sum_{a'_{0 \to T-1}}^{\mathcal{A}^T} \sum_{z_{0 \to T}}^{\mathcal{Z}^{T+1}} q(z_0|h_0, i = 0) \prod_{t=0}^{T-1} q(a'_t|z_t, h_t, i = 0) q(z_{t+1}, o_{t+1}|z_t, h_t, a_t, i = 0).$$

By considering the case $a'_{0 \to T-1} = a_{0 \to T-1}$ in isolation, and by assuming probabilities are positive, we readily obtain our first bound,

$$\prod_{t=0}^{T-1} q(o_{t+1}|h_t, a_t, i = 1) \geq \prod_{t=0}^{T-1} q(a_t|h_t, i = 0) q(o_{t+1}|h_t, a_t, i = 0).$$

In order to obtain our second bound, we further isolate the cases $a'_0 \neq a_0$, then $a'_0 = a_0 \wedge a'_1 \neq a_1$, then $a'_0 = a_0 \wedge a'_1 = a_1 \wedge a'_2 \neq a_2$ and so on until $a'_{0 \to T-2} = a_{0 \to T-2} \wedge a'_{T-1} \neq a_{T-1}$, which yields

$$\prod_{t=0}^{T-1} q(o_{t+1}|h_t, a_t, i = 1) = \prod_{t=0}^{T-1} q(a_t|h_t, i = 0) q(o_{t+1}|h_t, a_t, i = 0)$$

$$+ \sum_{z_{0 \to T}}^{\mathcal{Z}^{T+1}} q(z_0|h_0, i = 0) \left(1 - q(a_0|z_0, h_0, i = 0)\right) \prod_{t=0}^{T-1} q(z_{t+1}, o_{t+1}|z_t, h_t, a_t, i = 0)$$

$$+ \sum_{K=0}^{T-2} \sum_{z_{0 \to T}}^{\mathcal{Z}^{T+1}} q(z_0|h_0, i = 0) \prod_{t=0}^{K} q(a_t, z_{t+1}, o_{t+1}|z_t, h_t, i = 0) \left(1 - q(a_K|z_K, h_K, i = 0)\right)$$

$$\prod_{t=K+1}^{T-1} q(z_{t+1}, o_{t+1}|z_t, h_t, a_t, i = 0).$$

Then by assuming probabilities are upper bounded by 1, we obtain

$$\prod_{t=0}^{T-1} q(o_{t+1}|h_t, a_t, i = 1) \leq \prod_{t=0}^{T-1} q(a_t|h_t, i = 0) q(o_{t+1}|h_t, a_t, i = 0) + 1 - q(a_0|h_0, i = 0)$$

$$+ \sum_{K=0}^{T-2} \prod_{t=0}^{K} q(o_{t+1}|h_t, a_t, i = 0) \left( \prod_{t=0}^{K-1} q(a_t|h_t, i = 0) - \prod_{t=0}^{K} q(a_t|h_t, i = 0) \right)$$

$$\leq \prod_{t=0}^{T-1} q(a_t|h_t, i = 0) q(o_{t+1}|h_t, a_t, i = 0) + 1 - \prod_{t=0}^{T-1} q(a_t|h_t, i = 0).$$

Finally, with $\hat{q}$ solution of (3) and $|\mathcal{D}_{obs}| \to \infty$ we have that $D_{\mathrm{KL}}(p(\tau|i=0)\|\hat{q}(\tau|i=0)) = 0$, and thus $\hat{q}(a_t|h_t, i=0) = p(a_t|h_t, i=0)$ and $\hat{q}(o_{t+1}|h_t, a_t, i=0) = p(o_{t+1}|h_t, a_t, i=0)$, which shows the desired result. □

**Corollary 4.3.** *For any $\mathcal{D}_{int}$, the estimator $\hat{q}(o_{t+1}|o_{0\to t}, do(a_{0\to t}))$ recovered after solving (3) with $|\mathcal{D}_{obs}| \to \infty$ offers strictly better generalization guarantees than the one with $|\mathcal{D}_{obs}| = 0$.*

*Proof.* There exists at least one history-action couple $(h_{T-1}, a_{T-1})$, $T \geq 1$, that has non-zero probability in the observational regime. This ensures that there exists a value $o_T$ for which $\prod_{t=0}^{T-1} p(a_t|h_t, i=0)p(o_{t+1}|h_t, a_t, i=0)$ is strictly positive, which in turn ensures $\hat{q}(o_{T+1}|h_T, a_T, i=1) > 0$ (Theorem 4.2). As a result, the family of models $\{q(o_{t+1}|h_t, a_t, i=1) \mid q \in \mathcal{Q}, q(\tau|i=0) = p(\tau|i=0)\}$ is a strict subset of the unrestricted family $\{q(o_{t+1}|h_t, a_t, i=1) \mid q \in \mathcal{Q}\}$, and thus offers strictly better generalization guarantees. □

# D  Experimental details

The code for reproducing our experiments is made available online[5].

We perform experiments on three synthetic toy problems: the *door* problem described earlier (Figure 6), the classical *tiger* problem from the literature [Cassandra et al., 1994], and a 5x5 *gridworld* problem inspired from Alt et al. [2020].

**Data**  To assess the performance of our method, we consider a large observational dataset $\mathcal{D}_{prv}$ of fixed size (512 samples for *door*, 8192 samples for *tiger* and *gridworld*), and an interventional dataset $\mathcal{D}_{std}$ of varying size, ranging on an exponential scale from 4 to $|\mathcal{D}_{prv}|$.

**Baselines**  We compare the performance of the transition model $\hat{q}$ recovered in three different settings: *no obs*, when only interventional data ($\mathcal{D} = \mathcal{D}_{std}$) is used for training; *naive*, when observational data is naively combined with interventional data as if there was no confounding ($\mathcal{D} = \mathcal{D}_{std} \cup \{(\tau, 1)|(\tau, i) \in \mathcal{D}_{prv})\}$); and *augmented*, our proposed method ($\mathcal{D} = \mathcal{D}_{std} \cup \mathcal{D}_{prv}$). Note that the only difference between each of those settings is the training dataset, all other aspects (learning procedure, model architecture, loss function) begin the same.

**Training**  In all our experiments we use a tabular model for $\hat{q}$, that is, we use discrete probability tables for each building blocs of the transition model, $q(z_0)$, $q(o_t|z_t)$, $q(z_{t+1}|z_t, a_t)$, and $q(a_t|h_t, z_t, i=0)$. We use a latent space $|\mathcal{Z}|$ of size 32, 32 and 128 respectively for each toy problem, while the true latent space $|\mathcal{S}|$ is of size 3, 6 and 42. We train $\hat{q}$ by directly minimizing the negative log likelihood (3) via gradient descent. We use the Adam optimizer [Kingma and Ba, 2015] with a learning rate of $10^{-2}$, and train for 500 epochs consisting of 50 gradient descent steps with minibatches of size 32. We divide the learning rate by 10 after 10 epochs without loss improvement (reduce on plateau), and we stop training after 20 epochs without improvement (early stopping). In the *door* experiment we derive the optimal policy $\hat{\pi}^\star$ exactly, while in the *tiger* and *gridworld* experiments we train a "dreamer" RL agent on imaginary samples $\tau \sim \hat{q}(\tau|i=1)$ obtained from the model, using the belief states $\hat{q}(s_t|h_t)$ as features. We use a simple Actor-Critic algorithm for training, and our agents consist of a simple MLP with one hidden layers for both the critic and the policy parts. RL agents are trained until convergence or with a maximum number of 1000 epochs, with a learning rate of $10^{-2}$, a discount factor $\gamma = 0.9$ and a batch size of 8.

**JS divergence**  To evaluate the general quality of the recovered transition models, we compute the expected Jensen-Shannon divergence between the learned $\hat{q}(o_{t+1}|h_t, i=1)$ and the true

---

[5]https://supplementary.materials/disclosed.after.acceptance

$p(o_{t+1}|h_t, i=1)$, over transitions generated using a uniformly random policy $\pi_{rand}$,

$$\frac{1}{2}\mathbb{E}_{\tau \sim p_{init}, p_{trans}, p_{obs}, \pi_{rand}} \left[ \log \frac{p(o_0)}{m(o_0)} + \sum_{t=1}^{|\tau|} \log \frac{p(o_{t+1}|h_t, i=1)}{m(o_{t+1}|h_t, i=1)} \right]$$

$$+ \frac{1}{2}\mathbb{E}_{\tau \sim \hat{q}_{init}, \hat{q}_{trans}, \hat{q}_{obs}, \pi_{rand}} \left[ \log \frac{\hat{q}(o_0)}{m(o_0)} + \sum_{t=1}^{|\tau|} \log \frac{\hat{q}(o_{t+1}|h_t, i=1)}{m(o_{t+1}|h_t, i=1)} \right],$$

where $m(.) = \frac{1}{2}\left(\hat{q}(.) + p(.)\right)$. In the first experiment we compute the JS exactly, while in the second experiment we use a stochastic approximation over 100 trajectories $\tau$ to estimate each of the expectation terms in the JS empirically.

**Reward.** To evaluate quality of the recovered transition models for solving the original RL task, that is, maximizing the expected long-term reward, we evaluate the policy $\hat{\pi}^{\star}$, obtained after planning with the recovered model $\hat{q}$, on the true environment $p$,

$$\mathbb{E}_{\tau \sim p_{init}, p_{trans}, p_{obs}, \hat{\pi}^{\star}} \left[ \sum_{t=0}^{|\tau|} R(o_t) \right].$$

In the first experiment we compute this expectation exactly, while in the second experiment we use a stochastic approximation using 100 trajectories $\tau$.

## E   Complete empirical results

### E.1   Door experiment

The *door* experiment (Figure 6) corresponds to a simple binary bandit setting, that is, a specific POMDP with horizon $H = 1$. The observation space is of size $|\mathcal{O}| = 0$, since the learning agent receives no observation, and the hidden state space is of minimal size $|\mathcal{S}| = 3$ to encode both the initial light color and the reward obtained afterwards. The bandit dynamics are described in Table 1.

| *light* | |
|---|---|
| red | green |
| 0.6 | 0.4 |

$p(light)$

| *light* | *button* | *door* closed | *door* open |
|---|---|---|---|
| red | A | 0.0 | 1.0 |
| red | B | 1.0 | 0.0 |
| green | A | 1.0 | 0.0 |
| green | B | 0.0 | 1.0 |

$p(door|light, button)$

Table 1: Probability tables for our *door* bandit problem.

We repeat the *door* experiment in six different scenarios, corresponding to different privileged policies $\pi_{prv}$ ranging from a totally random agent to a perfectly good and a perfectly bad agent. Each time, we evaluate the performance of the *no obs*, *naive* and *augmented* approaches under different data regimes, by varying the sample size for both the observational data $\mathcal{D}_{prv}$ and the interventional data $\mathcal{D}_{std}$ in the range $(4, 8, 16, 32, 64, 128, 256, 512)$.

In each scenario, we report both the expected reward and the JS as heatmaps with $|\mathcal{D}_{std}|$ and $|\mathcal{D}_{prv}|$ in the $x$-axis and $y$-axis respectively, to highlight the combined effect of the sample sizes on each approach. We also provide as a heatmap the difference between our approach, *augmented*, and the two other approaches *no obs* and *naive*. We always plot the expected reward in the first row, and JS in the second row. As a remark, shades of green show gains in reward (the higher the better), while shades of purple show gains in JS (the lower the better).

Finally, we also present two plots which provide a focus on the data regime that corresponds to the largest number of observational data ($|\mathcal{D}_{prv}| = 512$), as in the main paper.

**Noisy Good Expert**   In the noisy good expert setting, the expert plays halfway between a perfect and a random policy. The diversity of its action leads to a good start for the *naive* model but the bias it contains is hard to overcome. In contrast, our method makes good use of the observational data from the start and is also able to correct the bias as interventional data come in, eventually converging towards the true transition model.

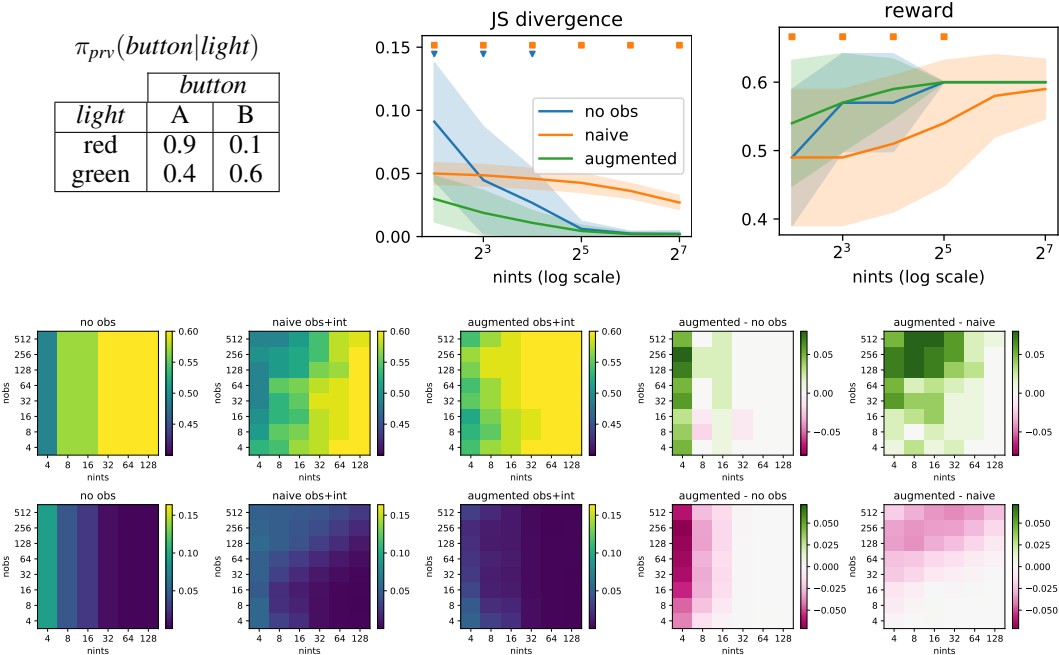

Figure 7: Noisy good expert setting. Heatmaps correspond respectively to the expected reward (top row, higher is better) and the JS divergence (bottom row, lower is better).

**Random Expert** A random policy naturally results in unconfounded observational data, since it does not exploits the privileged information. Hence, the *naive* approach is unbiased in this case, and actually makes the best use of the observational data. Our approach, *augmented*, exhibits an overall comparable performance, only slightly worse at times. We believe this can be explained by the additional complexity of our method which tries to disentangle a confounded regime in the data, and is not best suited to unconfounded data.

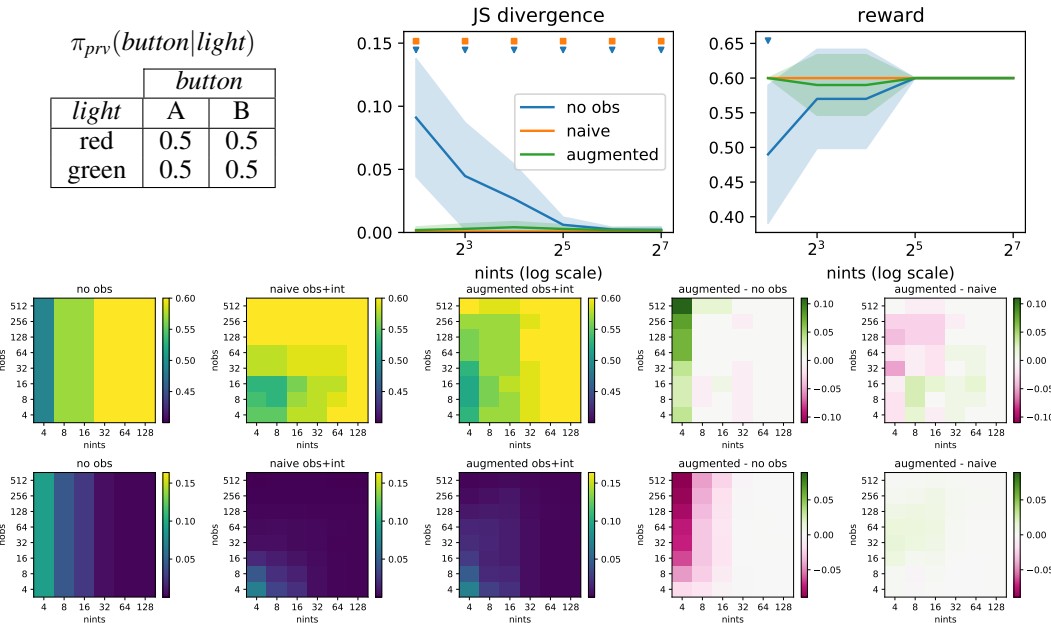

Figure 8: Random expert setting. Heatmaps correspond respectively to the expected reward (top row, higher is better) and the JS divergence (bottom row, lower is better).

**Perfectly Good Expert**    Observing a perfectly good expert playing in the *door* problem induces a strong bias, because every observed action always results in a positive reward. As such, the *naive* approach struggles to learn a good transition model. The bias however is quickly corrected by our *augmented* approach, which eventually converges to the true transition model faster than the *no obs* approach.

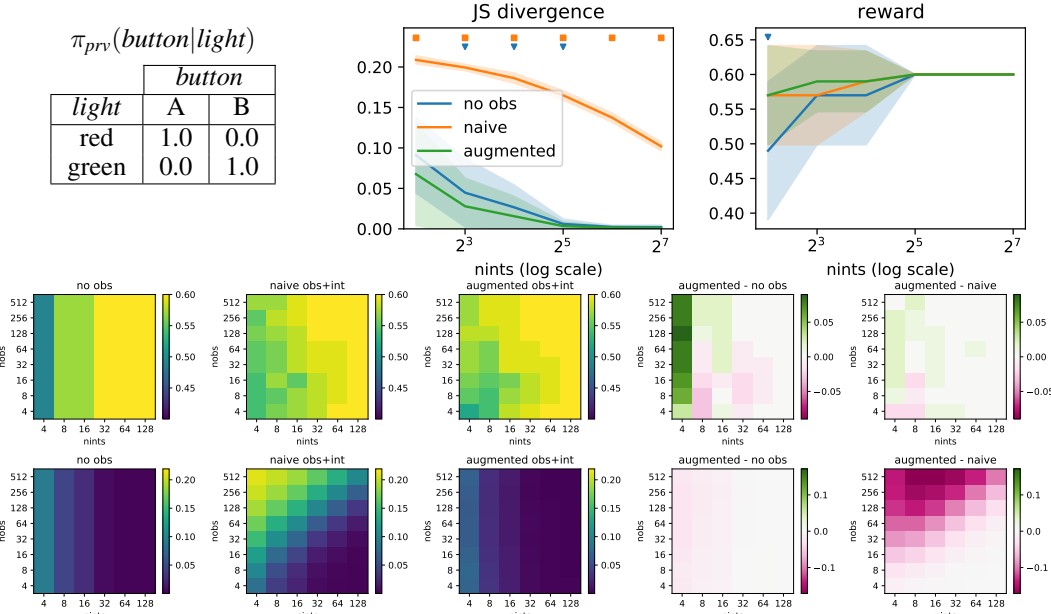

Figure 9: Perfectly good expert setting. Heatmaps correspond respectively to the expected reward (top row, higher is better) and the JS divergence (bottom row, lower is better).

**Perfectly Bad Expert**  Similarly to the previous setting, observing an expert that always chooses a bad action leads to a strong bias, as every action is associated to a low reward. The behaviour in terms of JS and reward is similar as well.

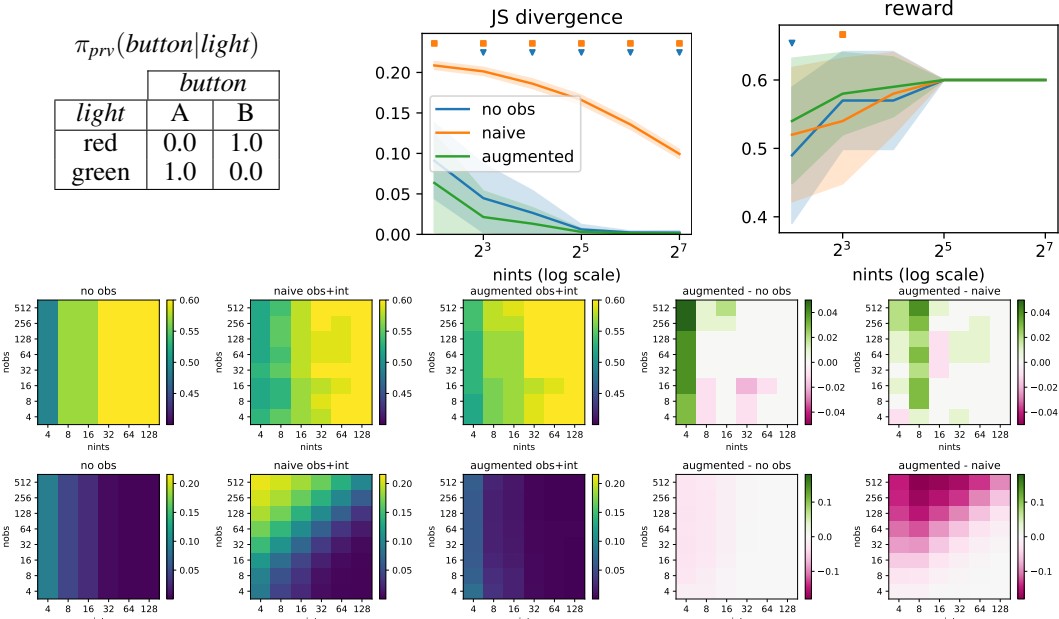

Figure 10: Perfectly bad expert setting. Heatmaps correspond respectively to the expected reward (top row, higher is better) and the JS divergence (bottom row, lower is better).

**Positively Biased Expert**    Here the expert's policy is considered as *positively biased* in the sense that the agent will only obtain a positive reward when playing button A (with 55% chances) and never by playing button B (0% chances). Because playing button A is actually the optimal policy, this strong bias has a positive effect on the reward for the *naive* approach. Hence, although worse in terms of JS than our approach, the *naive* approach always results in a very good policy in terms of reward. Our *augmented* approach, however, seems more conservative.

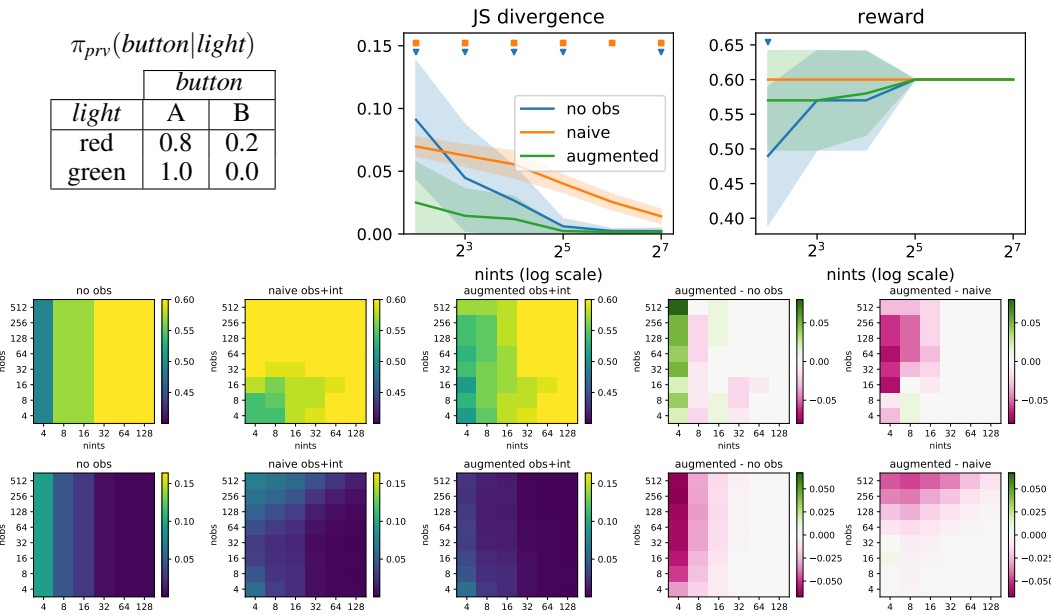

Figure 11: Positively biased expert setting. Heatmaps correspond respectively to the expected reward (top row, higher is better) and the JS divergence (bottom row, lower is better).

**Negatively Biased Expert**   In an analogous way, a negatively biased expert will overuse button A, leading to mixed feelings regarding this button, whereas it will always get a positive reward each time it uses button B. This leads to the opposite behavior as we had in the previous setting, with the *naive* approach always favoring the use of button B, and obtaining a bad performance in terms of reward. The *naive* approach only gets better when a lot of interventional data is combined with the biased observational data, while our *augmented* approach is able to overcome this pessimistic bias very early on, and converges faster than both *no obs* and *naive*.

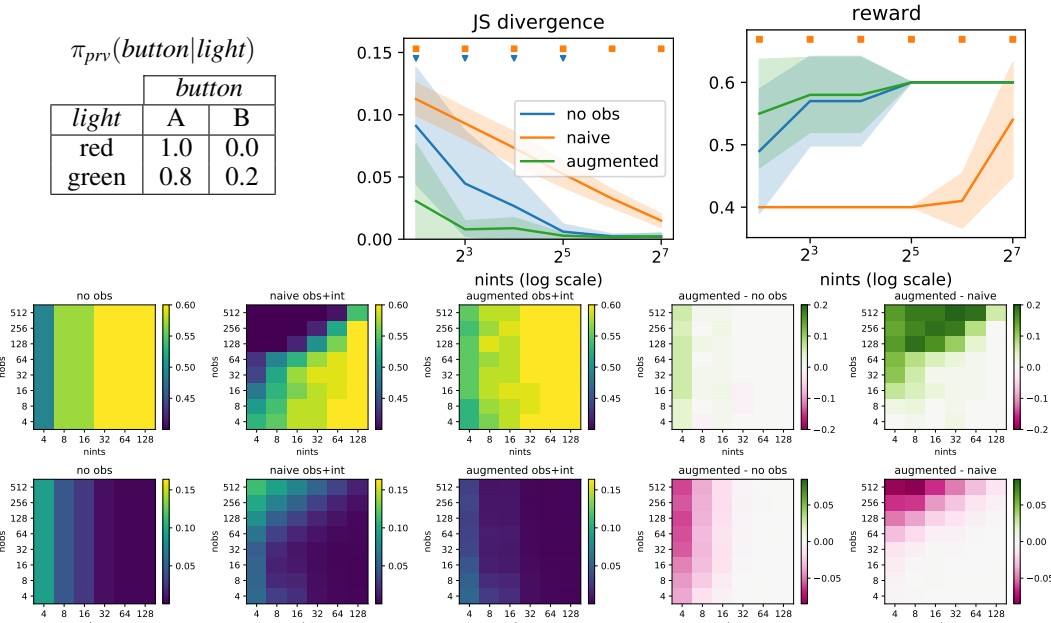

Figure 12: Pessimistic bias expert setting. Heatmaps correspond respectively to the expected reward (top row, higher is better) and the JS divergence (bottom row, lower is better).

### E.2 Tiger experiment

The *tiger* experiment corresponds a synthetic POMDP toy problem proposed by Cassandra et al. [1994]. In short, in this problem the agent stands in front of two doors to open, one of them having a tiger behind it (-100 reward), and the other one a treasure (+10 reward). The agent also gets a noisy observation of the system in the form of the roar from the tiger, which seems to originate from the correct door most of the time (85% chances) and the wrong door sometimes (15% chances). In order to reduce uncertainty the agent can listen to the tiger's roar again, at the cost of a small penalty (-1). We present the simplified POMDP dynamics in Table 2, and in our experiments we impose a fixed horizon of size $H = 50$. The observation space is of size $|\mathcal{O}| = 6$, to encode the roar location perceived by the agent and the obtained reward, $o_t = (roar_t, reward_t)$, and the hidden state space is of minimal size $|\mathcal{S}| = 6$ to encode both the tiger position and the reward obtained at each time step, $s_t = (tiger_t, reward_t)$.

| $tiger_0$ | |
|------|-------|
| left | right |
| 0.5 | 0.5 |

$p(tiger_0)$

| | $roar_t$ | |
|-----------|------|-------|
| $tiger_t$ | left | right |
| left | 0.85 | 0.15 |
| right | 0.15 | 0.85 |

$p(roar_t|tiger_t)$

| | | $tiger_{t+1}$ | |
|-----------|------------|------|-------|
| $tiger_t$ | $action_t$ | left | right |
| | listen | 1.0 | 0.0 |
| left | open left | 0.5 | 0.5 |
| | open right | 0.5 | 0.5 |
| | listen | 0.0 | 1.0 |
| right | open left | 0.5 | 0.5 |
| | open right | 0.5 | 0.5 |

$p(tiger_{t+1}|tiger_t, action_t)$

| | | $reward_{t+1}$ | | |
|-----------|------------|-----|------|-----|
| $tiger_t$ | $action_t$ | -1 | -100 | +10 |
| | listen | 1.0 | 0.0 | 0.0 |
| left | open left | 0.0 | 1.0 | 0.0 |
| | open right | 0.0 | 0.0 | 1.0 |
| | listen | 1.0 | 0.0 | 0.0 |
| right | open left | 0.0 | 0.0 | 1.0 |
| | open right | 0.0 | 1.0 | 0.0 |

$p(reward_{t+1}|tiger_t, action_t)$

Table 2: Probability tables for the *tiger* problem.

For the tiger experiment we consider four different privileged policies $\pi_{prv}$ for the observed agent. We then evaluate the performance of the *no obs*, *naive* and *augmented* approaches under different data regimes, by keeping the observational data fixed to $|\mathcal{D}_{prv}| = 8192$ while varying the varying the number of interventional data for $\mathcal{D}_{std}$ in the range $(4, 8, 16, 32, 64, 128, 256, 512, 1024, 2048, 4096, 8192)$.

**Noisy Good Expert**   In this scenario the privileged expert adopts a policy that plays the optimal action most of the time (open the treasure door), but also sometimes decides to just listen or to open the wrong door. As can be seen, in this scenario our *augmented* method makes the best use of the observational data, and is significantly better than both the *no obs* and *naive* approaches in the low-sample regime, both in terms of quality of the estimated transition model and obtained reward.

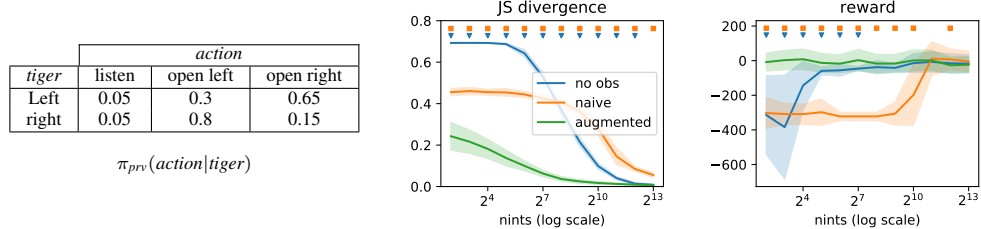

| | action | | |
|---|---|---|---|
| *tiger* | listen | open left | open right |
| Left | 0.05 | 0.3 | 0.65 |
| right | 0.05 | 0.8 | 0.15 |

$\pi_{prv}(action|tiger)$

Figure 13: Noisy good agent.

**Random Expert**   In the random scenario there is no confounding, and observational data can be safely mixed with interventional data. The *naive* approach thus does not suffer from any bias, and in fact is the one that converges the fastest to the optimal transition model and policy. Our method, while it manages to leverage the observational data to converge faster than *no obs*, suffers from a worse performance than *naive* in the low sample regime, most likely because it tries to recover a spurious confounding variable to distinguish the observational and interventional regimes, when none actually exists.

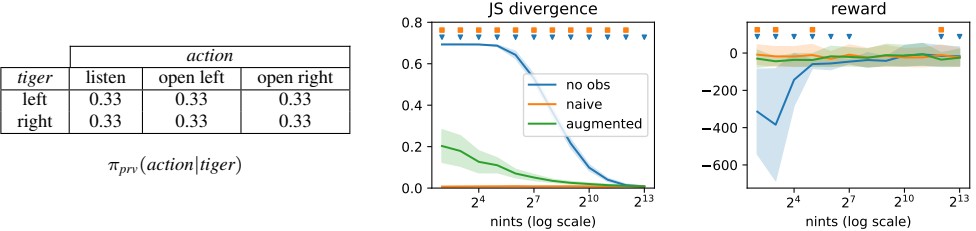

| | action | | |
|---|---|---|---|
| *tiger* | listen | open left | open right |
| left | 0.33 | 0.33 | 0.33 |
| right | 0.33 | 0.33 | 0.33 |

$\pi_{prv}(action|tiger)$

Figure 14: Random agent.

**Very Good Expert**  Here the privileged expert never opens the wrong door, and thus never receives the very penalizing -100 reward. As a result the *naive* approach seems to be overly optimistic with respect to the action of opening a door, which strongly affects the expected reward it obtains in the true environment. While our *augmented* approach seems also to suffer from this bias in the very low sample regime (as can be seen on the reward plot), overall the quality of the recovered transition model is still superior to both other approaches, and converges faster to the true transition model.

|  | *action* | | |
|---|---|---|---|
| *tiger* | listen | open left | open right |
| left | 0.05 | 0.0 | 0.95 |
| right | 0.05 | 0.95 | 0.0 |

$\pi_{prv}(action|tiger)$

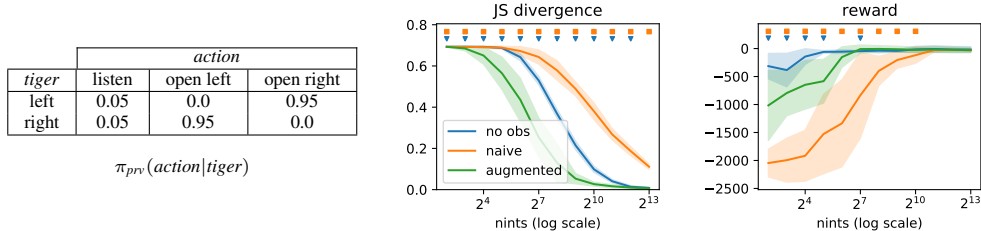

Figure 15: Very good agent.

**Very Bad Expert**  Here the privileged expert never opens the correct door, and thus never receives a positive reward (+10). As a result, the *naive* approach seems to be very conservative, and prefers not to take any chances opening a door. It turns out that this strategy is not too bad in terms of reward (always listening yields a -51 total reward), and as such this causal bias seems to positively affect the performance of the *naive* approach in the low sample regime, but prevents it from obtaining a better policy in the high sample regime too. Our *augmented* method, on the other hand, is able to escape this overly conservative strategy earlier on, and converges to a good-performing policy faster than both other approaches.

|  | *action* | | |
|---|---|---|---|
| *tiger* | listen | open left | open right |
| left | 0.05 | 0.95 | 0.0 |
| right | 0.05 | 0.0 | 0.95 |

$\pi_{prv}(action|tiger)$

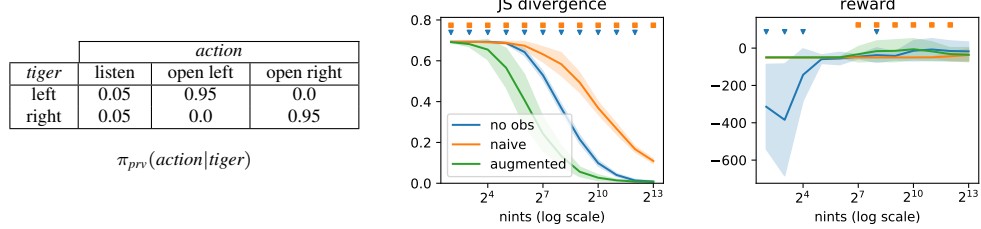

Figure 16: Very bad agent.

### E.3 Gridworld experiment

The *gridworld* experiment, represented in Figure 17, is inspired from [Alt et al., 2020]. It consists in a small 5x5 grid where the agent starts on the top-left corner, and tries to get to a target placed on the bottom side behind a large wall. The agent can use five actions: *top*, *right*, *bottom*, *left* and *idle*, and only receives a noisy signal about its current position. At each time step, the agent's position is revealed with 20% chances, and remains completely hidden otherwise. In addition, the agent's actions only have a stochastic effect, i.e., the agent moves into the desired direction with 50% chances, and otherwise slips at random to one of the 5 adjacent tiles or current tile. In case the agent would bump into a wall, it simply remains at its current position. The observation space is of size $|\mathcal{O}| = 44$, to encode both the agent's location (or the indication that the location is hidden) and the reward, and the hidden state space is of size $|\mathcal{S}| = 21$ to encode the agent's location. In this experiment we impose a fixed horizon of size $H = 20$.

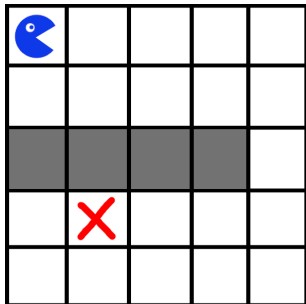

Figure 17: The gridworld problem

For the gridworld experiment we consider a single policy $\pi_{prv}$ for the privileged agent, who acts optimally (shortest path from current location to target). We then evaluate the performance of the *no obs*, *naive* and *augmented* approaches under different data regimes, by keeping the observational data fixed to $|\mathcal{D}_{prv}| = 8192$ while varying the varying the number of interventional data for $\mathcal{D}_{std}$ in the range $(4, 8, 16, 32, 64, 128, 256, 512, 1024, 2048, 4096, 8192)$.

**Very Good Expert** In this scenario the privileged agent adopts a perfect policy, and always chooses an action leading to the shortest path towards the target. As can be seen, here again our *augmented* method makes the best use of the observational data, and converges faster than both the *no obs* and the *naive* approaches for recovering the true transition model. This improvement in the transition model also translates into an improvement in terms of the learned policy, which starts converging towards high reward values with fewer samples ($2^7$) than both *no obs* and *naive* ($2^9$).

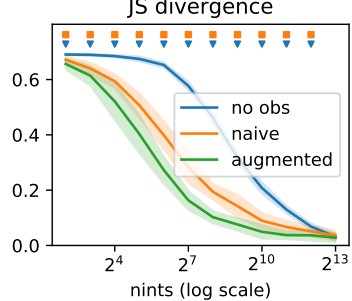 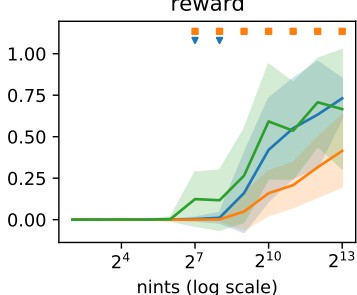

Figure 18: Perfect agent.

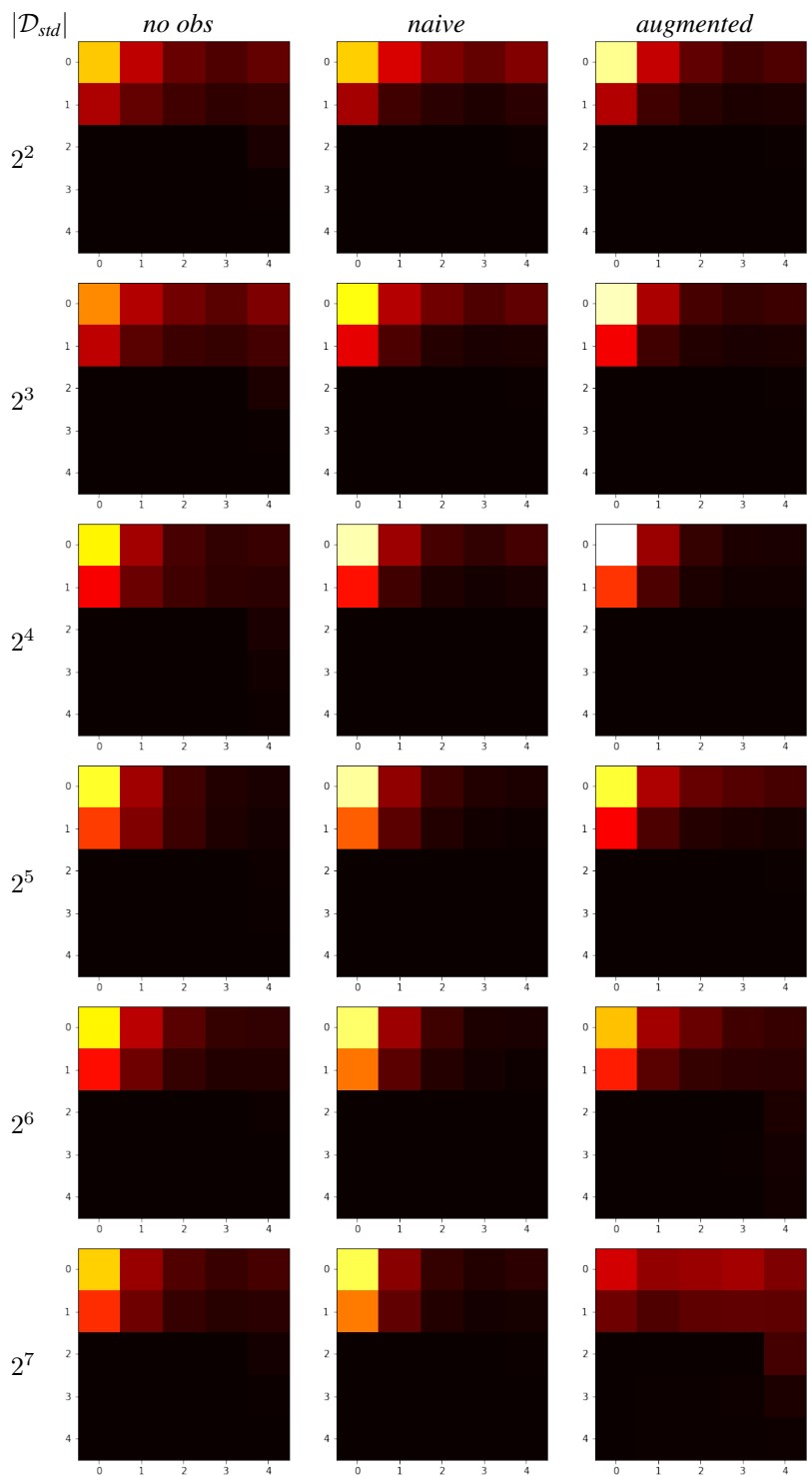

Figure 19: Average heat-maps over 100 episodes $\times$ 10 seeds, of the tiles visited by each trained agent (*no obs*, *naive*, *augmented*) for different interventional data sizes ($2^2$, $2^3$, $2^4$, $2^5$, $2^6$, $2^7$). The *augmented* approach is the fastest (in terms of interventional data) to learn how to properly escape the top part of the maze through tile (4, 2), and then move towards the treasure on tile (1, 3).

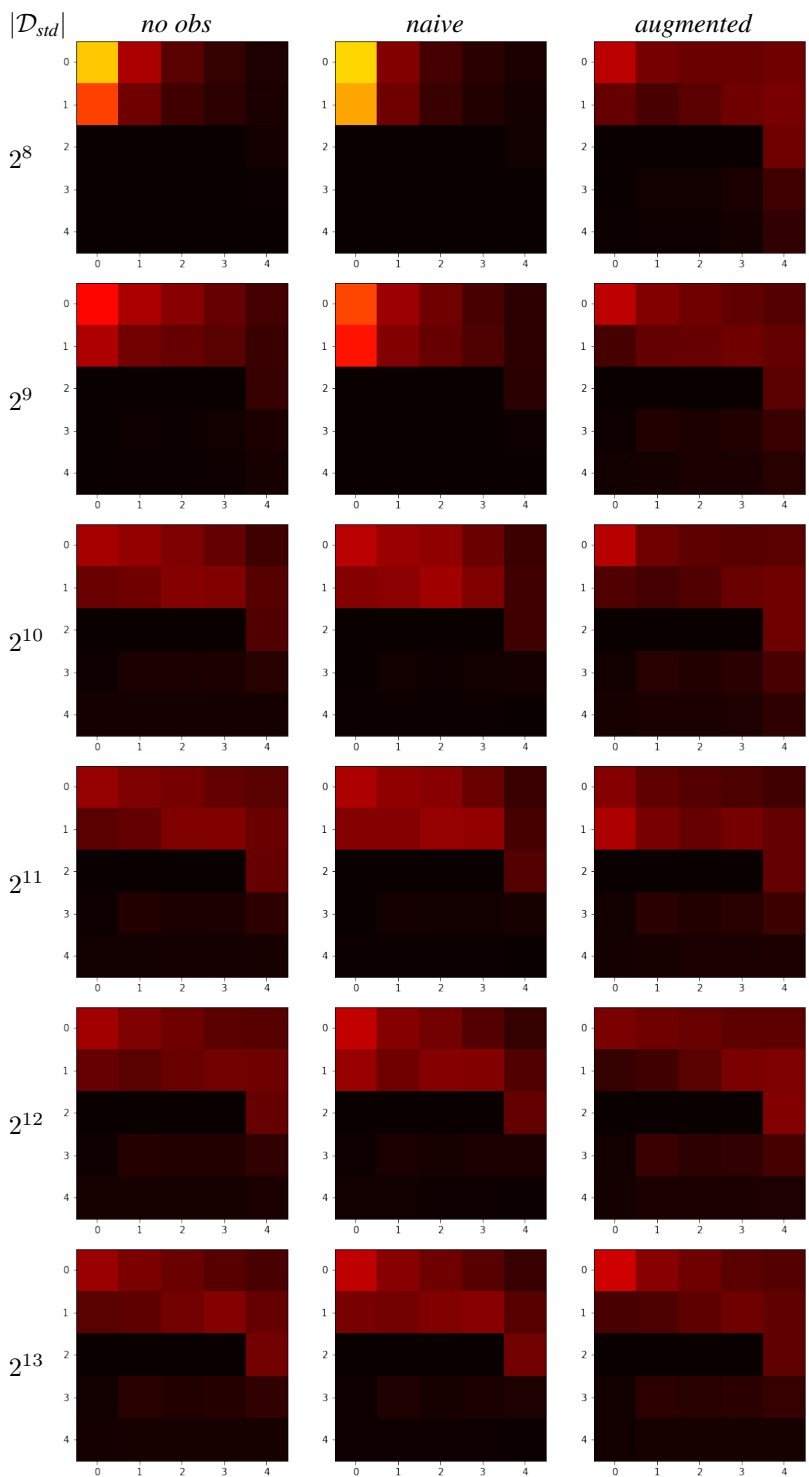

Figure 20: Average heat-maps over 100 episodes $\times$ 10 seeds, of the tiles visited by each trained agent (*no obs*, *naive*, *augmented*) for different interventional data sizes ($2^8$, $2^9$, $2^{10}$, $2^{11}$, $2^{12}$, $2^{13}$). The *augmented* approach is the fastest (in terms of interventional data) to learn how to properly escape the top part of the maze through tile (4, 2), and then move towards the treasure on tile (1, 3).

