# OpenReview forum: "Using Confounded Data in Offline RL"
_NeurIPS.cc/2022/Workshop/Offline_RL — Offline RL Workshop NeurIPS 2022_

### Official Review · Reviewer_QX4x · 2022-10-12
**Interesting ideas however very unclear how it all fits together practically**

**Rating:** 5
**Confidence:** 5

**Review:**

This paper proposes a way to improve a learned policy by jointly accounting for online and offline data streams. It is proposed that by treating the integration, when learning in a model-based fashion, as a causal inference problem the learned policy obtains some generalization guarantees as well as added efficiency in attaining optimal performance.

While the claims made in the paper are appealing, I found this version of the work to be unclear and confusing. It's apparent that this submission is derived from a fuller paper. In the transition to the format required for submission significant gaps were created in the writing and presentation of the work that greatly reduce the impact the intended evidence supporting the claims.

I'll try to exhaustively list the areas that are unclear that keep me from confidently recommending this paper to be accepted. Next I'll highlight some papers that probably should be cited and compared to in future work. I'll then summarize with one overarching concern of the suitability of this paper for the workshop.

- There's little explanation or definition of the difference between interventional regime and observational regime. These terms seem to be used interchangeably with "online" and "offline" data. Giving a more direct description of where each regime comes from would greatly help clarify what is meant by each and how they are used.
- While I really like the distinctions made between standard and privileged POMDPs, I wish that the specific terms used here were more clearly integrated with the "interventional"/online and "observational"/offline regime designation. I think that it was assumed that using "expert" when describing the privileged data was sufficient, but it didn't complete the connection very well.
- A more clear connection (with formal language and definition--eg. with math) between the interventional regime and model-based RL is necessary.
- Specifying the risk of applying off-the-shelf RL to confounded data is warranted. Just vaguely referencing this risk (without the specific factors) is annoying, especially when there's been work along these lines! (e.g. Oberst and Sontag (ICML 2019); see below)
- It's not clear how the proposed two-phase learning process (Learning then Inference) is different from standard counterfactual inference.
- It's not clear how the estimators $q$ are decomposed into the different components --> particularly how each of these components are represented and learned!
- The training procedure is not at all specified and there is nothing explained to differentiate the proposed approach from the naive baseline. From the submitted paper, there doesn't seem to be any difference procedurally. Even after perusing the appendix, this difference is not at all apparent.

There's been tons of work connecting RL and Causal Inference and while I'll not exhaustively list everything, I will include several papers of note that are relevant and should be considered by the authors going forward.
  - "Deconfounding Reinforcement Learning in Observational Settings"; Lu, et al (2018 arxiv)
  - "Woulda, Coulda, Shoulda: Counterfactually-guided policy search"; Buesing, et al (2019 ICLR)
  - "Counterfactual Off-Policy Evaluation with Gumbel-Max Structural Causal Models"; Oberst and Sontag (2019 ICML)
  - "Causally correct partial models for RL"; Rezende, et al (2019 arxiv)
  - "Causality and batch RL: Complementary Approaches to Planning in Unknown Domains"; Bannon, et al (2020 arxiv)
  - "Counterfactual Credit Assignment in model-free RL"; Mesnard, et al (2021 ICML)
  - "Training a Resilient Q-Network against Observational Interference"; Yang, et al (2022 AAAI)
  - "Counterfactually guided policy transfer in clinical settings"; Killian, et al (2022 CHIL)

As a final note I did want to question the appropriate-ness of this paper for the Offline RL workshop. It's clear that there is some utility in investigating the use of offline data to improve a policy's performance. Especially in a model-based setting!!! The authors approach to posing this as a way to relieve issues with confounding is notable and really interesting. I do however look at the experiments relying so heavily on active data collection as a way to differentiate between approaches and actually learn a performant policy (excluding the oracle data used for the Tiger domain). Perhaps the connections to offline RL are more clear in how the proposed method is actually implemented and used. Since this wasn't made clear in the paper I do not have the ability to adequately advocate for its inclusion in the workshop.

---

### Official Review · Reviewer_EGnZ · 2022-10-14

**Rating:** 7
**Confidence:** 4

**Review:**

### Summary

The paper proposes to use confounding in offline RL by leveraging a causal model over states and actions. While confounding in the fully offline setting is known to hurt agent performance, mixing confounders with online non-confounding samples hints at sample-efficient learning. The paper leverages this insight in the model-based setting. A latent causal model is learned to represent initialization, transition and observation distributions using the do-calculus framework. The model is utilized in the augmented POMDP setting wherein interventional variables indicate potential confounding in the agent policy.

### Strengths

1. The paper is well written and easy to understand.
2. Introductory examples provided by the authors are a simple way to involve reader with causality.

### Weaknesses

1. My main concern has to do with the augmented POMDP setting. How is the introduction of interventional variables different from simply combining offline and new online data? While confounders are a meaningful way to inform agents about cause of their actions, a naive model-based approach fulfills these requirements as well. It would be interesting if the authors could draw comparisons with standard model-based RL and how causality would aid in addressing model uncertainty.
2. A thorough discussion of literature and past work has not been provided. I request the authors to add discussion on relevant related work on offline RL, model-based RL and causality in general.
3. This might be a minor comment, but the results for augmented method are not convincing enough. Compared to No Obsverations and Naive method, the augmented POMDP seldomly helps the agent improve its returns. On the other hand, augmented POMDP almost always improves JS divergence. My understanding is that the causal model is an accurate model of data but experiment scenarios possess an easy-to-learn reward function. If the authors could add a complex high-dimensional task then benefits might be more apparent.